# The effect of tranexamic acid on intraoperative blood loss in patients undergoing brain meningioma resections: Study protocol for a randomized controlled trial

**Haojie Yu‡, Minying Liu‡, Xingyue Zhang, Tingting Ma, Jingchao Yang, Yaru Wu, Jie Wang, Muhan Li, Juan Wang, Min Zeng, Liyong Zhang, Hailong Jin, Xiaoyuan Liu, Shu Li⊙*, Yuming Peng**

Department of Anesthesiology, Beijing Tiantan Hospital, Capital Medical University, Beijing, China

‡ HY and ML are contributed equally to the work as co-first authors on this work
* lishu@bjtth.org

**Data Availability Statement:** No datasets were generated or analysed during the current study. All

## Abstract

### Introduction

Tranexamic acid (TXA) has been proven to prevent thrombolysis and reduce bleeding and blood transfusion requirements in various surgical settings. However, the optimal dose of TXA that effectively reduce intraoperative bleeding and blood product infusion in patients undergoing neurosurgical resection of meningioma with a diameter $\geq$ 5 cm remains unclear.

### Methods

This is a single-center, randomized, double-blinded, paralleled-group controlled trial. Patients scheduled to receive elective tumor resection with meningioma diameter $\geq$ 5 cm will be randomly assigned the high-dose TXA group, the low-dose group, and the placebo. Patients in the high-dose TXA group will be administered with a loading dose of 20 mg/kg TXA followed by continuous infusion TXA at a rate of 5 mg/kg/h. In the low-dose group, patients will receive the same loading dose of TXA followed by a continuous infusion of normal saline. In the control group, patients will receive an identical volume of normal saline. The primary outcome is the estimated intraoperative blood loss calculated using the following formula: collected blood volume in the suction canister (mL)–the volume of flushing (mL) + the volume from the gauze tampon (mL). Secondary outcomes include calculated intraoperative blood loss, intraoperative coagulation function assessed using thromboelastogram (TEG), intraoperative cell salvage use, blood product infusion, and other safety outcomes.

### Discussion

Preclinical studies suggest that TXA could reduce intraoperative blood loss, yet the optimal dose was controversial. This study is one of the early studies to evaluate the impact of

relevant data from this study will be made available upon study completion.

**Funding:** The trial is supported by Beijing Municipal Administration of Hospitals Incubating Program (PX2022018). The funders had no role in study design, data collection and analysis, decision to publish, or preparation of the manuscript.

**Competing interests:** The authors have declared that no competing interests exist.

intraoperative different doses infusion of TXA on reducing blood loss in neurological meningioma patients.

## Trial registration

ClinicalTrials.gov, NCT05230381. Registered on February 8, 2022.

## Introduction

Meningiomas are the most common intracranial tumors, accounting for about 36.0% [1], of which huge meningiomas, defined as diameter $\geq$ 5 cm, account for about 11% in primary intracranial neoplasm [2]. Although surgical resection of huge meningiomas is the preferred treatment to improve neurological outcomes, patients with huge meningiomas are at high risk of massive bleeding due to the space-occupying effect on brain tissue, increased intracranial pressure, and compression of blood vessels. Patients undergoing meningioma resection were associated with excessive blood loss of more than 2000 mL and life-threatening hemodynamic instability [2–4]. Previously, a prospective randomized controlled trial reported the mean intraoperative blood loss in huge meningioma patients was 1124 mL [4]. Besides that, the tissue plasminogen activator in tumor and peritumoral brain tissues can induce fibrinolysis and contribute to intraoperative hemorrhage [5]. Massive intraoperative bleeding (defined as receiving at least 5 units of red blood cells within 1 day of surgery) was reported to be associated with increased odds of perioperative death (Odd ratio: 8.1, 95% confidence interval (CI): 3.9–17.0), prolonged duration of mechanical ventilation, ICU stays, and hospital stay in a large retrospective study [2, 6]. Therefore, blood management of huge meningioma resection is challenging [2, 7–10].

Several perioperative blood management strategies are used in neurosurgery settings, including preoperative embolization of feeding arteries, topical hemostatic agents, and intraoperative cell salvage [11, 12]. However, these techniques or hemostatic agents have potential side effects, adverse events, and with higher medical costs. In a retrospective observational study, Brandel et al. demonstrated that patients with meningioma receiving preoperative endovascular embolization have higher rates of cerebral edema (25.3% vs. 17.7%), post-hemorrhagic anemia or transfusion (21.8% vs. 13.8%, p = 0.0003) [3], even lead to postoperative hemorrhage, stroke, and cranial nerve palsies [13]. Intraoperative cell salvage was introduced in neurosurgery for its efficacy in reducing the need for allogeneic RBC transfusion, but microbiological contamination and distant metastasis were also reported [14, 15]. Excessive use of topical hemostatic agents (including gelatine sponge (Gelfoam), gelatine-thrombin matrix sealant (FloSeal), microfibrillar collagen (Avitene), oxidized regenerated cellulose (Surgicel), and fibrin sealants (Tisseel)) was mistakenly reported as abscess or tumor recurrence mimicking, thus interfere the postoperative evaluation [16]. In cases of huge meningioma with massive hemorrhage, besides the above-mentioned blood management techniques, allogeneic transfusion would be adopted, with risks of transfusion immunosuppression, hemolysis, postoperative infection, and transfusion-related lung injury [17–21]. It is generally accepted that a multidisciplinary hemorrhage management strategy should be utilized to reduce blood loss and improve the care of patients with severe hemorrhage [11, 12].

Tranexamic acid (TXA) is a synthetic derivative of the amino acid lysine [19, 22–24] that exerts its antifibrinolytic effect through the reversible blockade of lysine-binding sites on plasminogen molecules [22]. TXA was reported to prevent thrombolysis, reduce bleeding, and reduce blood transfusion rates in several surgical settings, including trauma, otolaryngology,

obstetrics, cardiac surgery, and orthopedics [25–35]. In 2016, Mebel et al. demonstrated that TXA was associated with the reduced perioperative infusion of allogeneic blood products in patients undergoing complex skull base surgery [36]. However, the heterogeneous population and inevitable defect in the study design impeded supporting the efficacy and safety of TXA in huge meningioma patients. On the other hand, albeit a randomized controlled trial showed intraoperative TXA infusion significantly reduced blood loss (830 mL vs. 1124 mL, p = 0.03) in meningioma patients [4], the conclusion was difficult to interpret due to different tumor volumes in the study population. Goobie and his colleagues attempted to investigate the efficacy of TXA in neurosurgery and conducted a non-inferiority randomized controlled trial of intraoperative infusion of different doses of TXA in children with craniosynostosis [37]. The results demonstrated that low-dose TXA was not less effective than high-dose. However, it was too arbitrary to draw the same conclusion on the meningioma population as craniosynostosis surgery is an extracerebral procedure without manipulating brain tissue and pathological lesions, releasing tissue plasminogen activator. Therefore, whether TXA can effectively reduce intraoperative blood loss in patients undergoing huge meningioma surgery without increasing perioperative adverse events remains unclear.

Therefore, we hypothesize that intraoperative TXA would effectively reduce intraoperative blood loss in supratentorial meningioma resection and propose to conduct a trial to compare the efficacy of different doses of intraoperative TXA in reducing intraoperative bleeding in patients undergoing huge meningioma resection.

## Methods

The protocol has been prepared and presented following the Standard Protocol Items: Recommendations for Interventional Trials [38], and all trial procedures are summarized in Table 1.

### Study design

A randomized, double-blinded, paralleled-group controlled study will be performed to investigate the effect of TXA on intraoperative blood loss in patients undergoing huge meningioma (diameter ≥ 5 cm) resection. Patients scheduled to perform elective huge meningioma (diameter ≥ 5 cm) resection will be enrolled from Beijing Tiantan Hospital, Capital Medical University, from 2022 to 2024. The study was registered on www.clinicaltrials.gov (NCT05230381) on February 8, 2022, and approved by the Ethics Committee of Beijing Tiantan Hospital, Capital Medical University (ky2022-075-02) following the criteria required by the Helsinki Declaration. Preoperative interviews will be conducted by trained research assistants. Patients will be informed of the study objectives, risks, and benefits. Written informed consent will be obtained from participants or his/her legal surrogate. The flowchart is briefly illustrated in Fig 1.

### Study population

**Inclusion criteria.** Patients scheduled to undergo elective huge supratentorial meningioma resection will be recruited for screening eligibility one day before surgery. Those who agree to participate must sign the informed consent form and be informed that they can withdraw from the study at any time if they wish. Inclusion criteria include:

1. Age between 18 and 65 years.

2. Preoperative computed tomography (CT) or magnetic resonance imaging (MRI) suggested meningioma no less than 5cm in diameter in any of three axes.[39, 40]

3. American Society of Anesthesiologists (ASA) physical status I to III.

**Table 1. Schedule of enrollment, intervention, and assessment.**

| | Study period | | | | | | |
| | Enrollment | Allocation | Post-allocation | | Follow-up | | |
| Time point | Pre-operation | after evaluation | Intraoperative | PACU | 24h after operation | Discharge (7 days after operation) | 180 days after operation |
|---|---|---|---|---|---|---|---|
| **Enrollment** | | | | | | | |
| Eligibility criteria assessment | √ | | | | | | |
| Informed consent | √ | | | | | | |
| Sociodemographic questionnaire | √ | | | | | | |
| Allocation | | √ | | | | | |
| **Interventions** | | | | | | | |
| High-dose TXA | | | √ | | | | |
| Low-dose TXA | | | √ | | | | |
| Placebo | | | √ | | | | |
| **Assessments** | | | | | | | |
| Glasgow Coma Scale | √ | | | | √ | √ | √ |
| Caprini scale | √ | | | | √ | | |
| Karnofsky Performance Status Scale | √ | | | | | √ | √ |
| Age-adjusted Charlson comorbidity score | √ | | | | | | |
| Estimated cumulative blood loss* | | | √ | | | | |
| Operative blood loss | | | √ | | | | |
| Coagulation function | | | √ | | | | |
| Cell salvage use and allogeneic blood products use | | | √ | | | | |
| Extent of meningioma resection | | | √ | | | | |
| Operative field surgical hemostasis score | | | √ | | | | |
| Hemostasis difficulty | | | √ | | | | |
| Epileptic seizures | | | √ | √ | √ | √ | √ |
| Other complications | | | √ | √ | √ | √ | √ |
| Transfusion volume and rate | | | √ | √ | √ | √ | |
| Observer's Assessment of Alertness/Sedation Scale | | | | √ | √ | √ | |
| All-cause mortality | | | | | √ | √ | √ |
| Length of hospital stays | | | | | | √ | |
| Length of ICU stays | | | | | | √ | |
| Total hospitalization cost | | | | | | √ | |

\* Primary outcome.

ICU, intensive care unit; TXA, tranexamic acid

Patients will be excluded if they are

1. Allergic to TXA,

2. With a history of thrombotic disease or chronic kidney disease (glomerular filtration rate <60 mL/min or albumin–creatinine ratio >30 mg/g) [41],

3. Under anticoagulant or antiplatelet therapy.

**Randomization and blinding.** Patients will be enrolled and randomized one day before surgery. Block randomization (block size of 6) will be applied based on a computer-generated

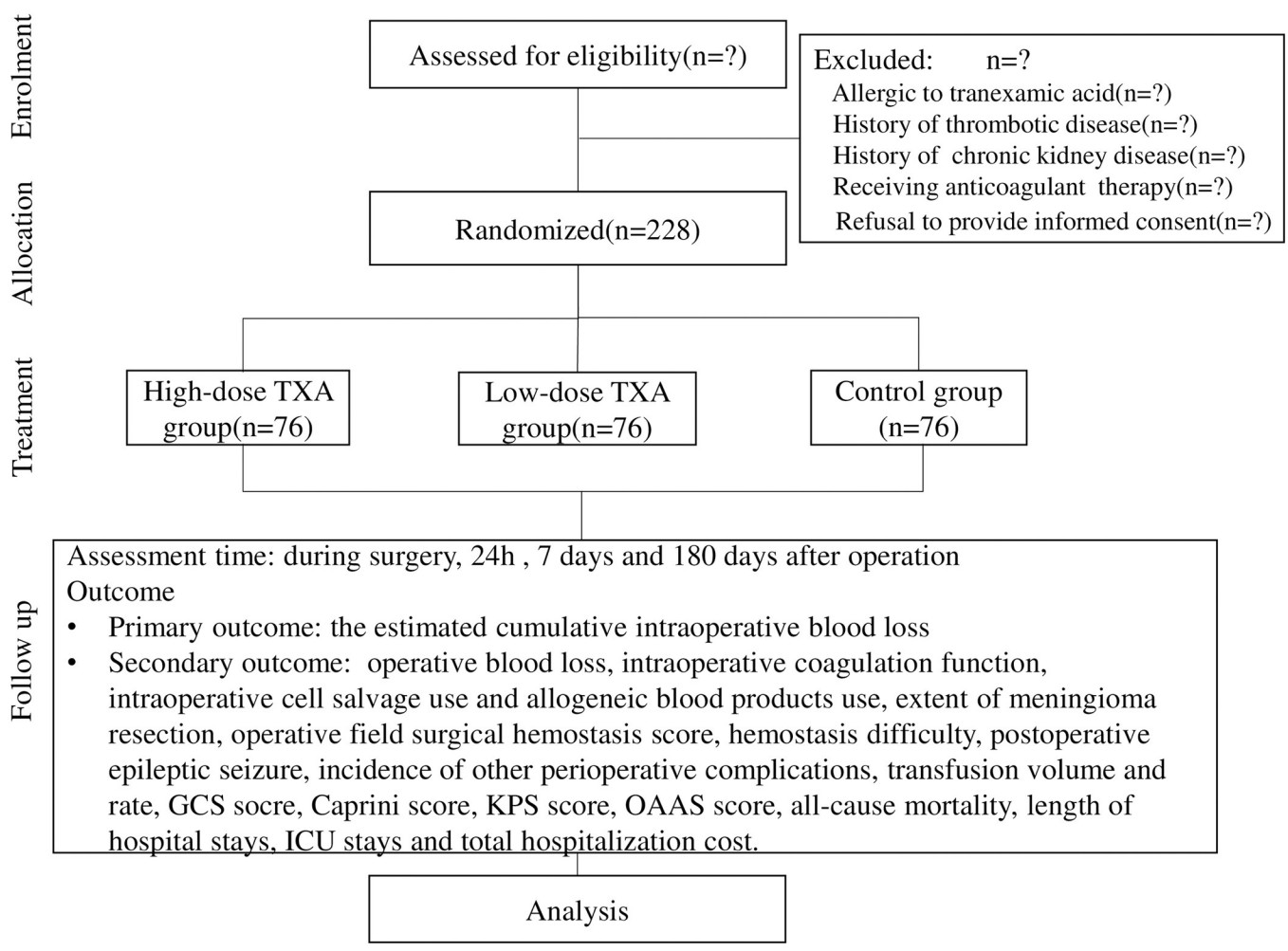

**Fig 1. Flowchart.**

table by an independent research assistant who has no role in the baseline assessment, anesthesia management, or follow-up. This independent research assistant will pack the allocation sequence with identical appearing, opaque, sealed, stapled envelopes and distribute them to the research nurse. The researcher nurse will open the envelopes and prepare the research drug based on the grouping. The loading infusion agents will be diluted in sterile, plastic, and opaque containers, and the continuous infusion agent will be prepared in a 50 mL syringe (all labeled "research agent"). Patients will be randomly assigned to the high-dose TXA group, low-dose TXA group, and placebo group with a 1:1:1 ratio.

Allocation will be concealed until the database lock. The responsible anesthesiologists, neurosurgeons, and outcome assessors will all be blinded to the allocation until the completion of the study analysis unless specific circumstances, including the occurrence of serious adverse events. The enrolled patients and his/her surrogates will also be blinded.

## Intervention and grouping

After endotracheal induction, patients in the high-dose TXA group will be administered a loading dose of TXA 20 mg/kg, followed by a continuous infusion of TXA at a rate of 5 mg/kg/hour until dura closure. In the low-dose group, patients will receive a loading dose of TXA 20

mg/kg followed by a continuous infusion of normal saline. In the control group, patients will receive an identical volume of normal saline in the same setting.

## Standard anesthesia management

Preoperative standard monitoring, including electrocardiograph (ECG), pulse oxygen saturation, non-invasive blood pressure (NIBP), heart rate (HR), bispectral index (BIS), and skin temperature will be initiated immediately after entering the operation room. Continuous invasive arterial pressure, end-tidal carbon dioxide ($ETCO_2$), urine output, and minimum alveolar concentration (MAC) of the inhalation agent will be monitored after anesthesia induction.

All patients will be premedicated with midazolam (0.03~0.05 mg/kg) intravenously 5 min before anesthesia induction. Propofol (1~2 mg/kg) or etomidate (0.2 mg/kg), rocuronium (0.9 mg/kg) or cisatracurium (0.2 mg/kg), and sufentanil (0.2~0.4 μg/kg) will be administered for anesthesia induction. After anesthesia induction and endotracheal intubation, mechanical ventilation will be performed to maintain normocapnia ($ETCO_2$ between 35 and 45 mmHg), at a tidal volume of 6 to 8 mL/kg, respiratory rate of 12 to 15/min, inspiratory/expiratory ratio of 1:2, 40% ~ 60% inspired oxygen fraction at a flow rate of 1~2 L/min. Anesthesia will be then maintained with combined intravenous anesthesia and inhalational anesthesia. Inhalation agent will be maintained at 0.4~0.5 MAC with remifentanil (0.05~0.2 μg/kg/min), and propofol (3~8 mg/kg/hour), to maintain the BIS value between 40 and 50. Sufentanil and muscle relaxant will be supplemented as needed.

Intraoperative mean arterial pressure will be maintained between ± 30% of the baseline value. The invasive arterial blood pressure monitoring will be performed through an A-line placed in the dorsal pedal arteries or radial artery to perform artery blood gas analysis (GAS) and thromboelastogram (TEG) before incision and after dura closure. Cell salvage will be adopted when the estimated blood loss is > 500mL [42]. Allogeneic erythrocytes, fresh frozen plasma, platelets, prothrombin complex concentrate, and cryoprecipitate will be infused according to the blood transfusion guideline [43, 44]. Fluid input and output will also be closely monitored and recorded.

Patients will receive postoperative patient-controlled intravenous analgesia treatment. The postoperative anti-seizure medication would be initiated according to the guideline [45]. Other administered drugs in the post-anesthesia care unit (PACU) or intensive care unit (ICU) will also be recorded.

## Data collection and measurement

After obtaining informed consent, an independent research assistant will initiate baseline information collection one day before surgery. Demographics, comorbidities, medical history, medication history, supplementary examination, and preoperative assessment will be measured and collected. Preoperative assessment includes Glasgow Coma Scale, Caprini scale, Karnofsky Performance Status Scale, Age-adjusted Charlson comorbidity score, and ASA physical status score. All patients will receive preoperative brain imaging to assess the tumor size, location and site, sinus involvement, and edema before surgery. Tumor size will be measured in the sagittal, coronal, and axial planes. To depict sinus involvement, contrast-enhanced magnetic resonance venography (CEMRV) will also be performed. The physiological parameters, coagulation assessed by TEG, the total doses of anesthetics, arterial blood gas analysis, and blood transfusion will be recorded by anesthesiologists through a designed data collection table. The Simpson grading system will describe the extent of meningioma resection [46]. The surgical grade of ooze will be evaluated using the previously reported scale [47], and hemostasis difficulty will be evaluated using the visual analogue scale (VAS) by neurosurgeons. Patients

will receive a postoperative brain CT scan within 12 hours postoperatively to detect intracranial hematoma, cerebral edema, hydrocephalus, newly onset stroke, and cerebral herniation. The functional status will be assessed using the Karnofsky Performance Status (KPS) at admission, 7 days, and 180 days after surgery. All-cause mortality and postoperative complications (defined in S1 Table) will also be recorded. Long-term follow-up will be performed through a remote video interview or telephone. Among them, seizures will be evaluated by neurosurgeons or neurologists using the International League Against Epilepsy (ILAE) operational definition of epilepsy [48]. To minimize loss of follow-up after discharge, study coordinators will contact the participants or their surrogates every three months.

## Outcome and safety measures

The study aims to observe the effect of TXA on intraoperative blood loss during the resection of a huge meningioma with a diameter $\geq$ 5 cm.

## Primary outcome

The primary outcome is the estimated cumulative intraoperative blood loss, which is calculated using the formula: collected blood volume in the suction canister (mL)–the volume of flushing (mL) + the volume from the gauze tampon (mL) [49]. For patients receiving cell salvage treatment, the estimated cell salvage efficiency is 50%, and the estimated blood loss is calculated using the formula: [(250mL/bowl) * number of bowls)/50% + volume from gauze tampon [43].

## Secondary outcome

The secondary outcomes include other efficacy parameters and safety outcomes.

1. Intraoperative calculated blood loss is assessed by the formula [50, 51]:
   Intraoperative calculated blood loss (CBL) = estimated blood volume $\times$ ($Hct_i$–$Hct_f$) + transfused RBC volume] / $Hct_{mean}$
   Estimated blood volume for women = $[\text{weight(kg)}^{0.425} \times \text{height(cm)}^{0.725}] \times 0.007184 \times 2217 + \text{age(years)} \times 106$
   Estimated blood volume for men = $[\text{weight(kg)}^{0.425} \times \text{height(cm)}^{0.725}] \times 0.007184 \times 3,064 - 825$
   where estimated blood volume is determined using the International Council For Standardization In Haematology (ICSH) formula [52]. $Hct_i$ is the initial hematocrit at the beginning of surgery, $Hct_f$ is the final hematocrit at closure, and $Hct_{mean}$ is the mean hematocrit (between initial and final).

2. Intraoperative coagulation function monitoring assessed by rapid TEG.

3. Intraoperative cell salvage use and allogeneic blood products use. Cell salvage will be used when estimated blood loss is >500 mL and intraoperative pathology consultation indicating World Health Organization classification $\leq$II [42]. Allogeneic erythrocyte transfusion will be initiated for patients with hemoglobin lower than 70 g/L or lower than 80 g/L in those with cardiovascular disease, acute ischemic stroke, or acute intraoperative bleeding [43]. Other blood products will be infused according to the blood transfusion guideline [44].

4. Simpson grading system for the extent of meningioma resection, in which grade 1 indicates total removal with the attached dura and involved abnormal bone. The higher the grade, the higher rate of residual tumor or infiltrated dura mater [46].

5. Operative field surgical hemostasis score using a 5-point scale, of which 0 indicates perfect hemostasis while 5 indicates poor hemostasis [47].

6. VAS score for hemostasis difficulty, where 0 indicates no difficulty in hemostasis while 10 indicates very difficult, almost impossible to achieve hemostasis.

7. A postoperative epileptic seizure, including early postoperative seizures, occurred during the first seven days after surgery and long-term seizures occur within six months after surgery. The subtype of seizure attack will be classified according to the ILAE definition [53].

8. Incidence of other perioperative complications (defined in S1 Table), including drug allergy, intraoperative hypoxemia, refractory hypotension, massive hemorrhage, deep vein thrombosis and pulmonary embolism, perioperative cerebral ischemia, intracranial hematoma, hydrocephalus, infection, acute renal injury, and acute myocardial injury.

9. Intraoperative and postoperative transfusion volume and rate within 7 days after the operation.

10. The Glasgow Coma Scale at 7 days and 180 days after the surgery.

11. The Caprini Scale at 24 h after the surgery.

12. The KPS Scale at admission, 7 days, and 180 days after the surgery.

13. The Observer's Assessment of Alertness/ Sedation Scale at PACU and 7 days after surgery.

14. All-cause mortality at day 7 and 180 days after surgery.

15. The length of hospital stays, ICU stays, and total hospital charge.

## Statistical analysis plan

The primary endpoint is estimated cumulative intraoperative blood loss and the analysis will be based on both the intention-to-treat (ITT) and the per-protocol (PP) approach. The ITT analysis will depend on the randomized population and the PP analysis will include participants who complete the treatment originally allocated. Descriptive statistics will be reported as means with standard deviation and medians with interquartile range for normally distributed data and skewed continuous data, respectively, and counts (percentage) for categorical data. We will compare normal distribution data with the Student's t-test and the Mann-Whitney U test for skewed data. The Chi-square test or Fisher's exact test will be used for categorical data. The primary outcome, estimated intraoperative blood loss, will be compared between groups using the Student't-test or the Mann-Whitney U test, and the multiple linear regression model or two-way analysis of covariance will be applied to account for potential confounders. The primary outcome will also be analyzed in the following subgroups: age, gender, and ASA physical status. The Bonferroni method will be used with an adjusted P value of 0.017 for between-group comparison, and the overall statistical significance will be declared at 0.05. Besides, missing data will be imputed using the worst-case scenarios and last observation carried forward method. All outcomes will be analyzed according to the statistical analysis plan with STATA (16.1, StataCorp LLC, College Station, TX).

## Sample size estimation

Sample size calculation was performed using the PASS 11 software (NCSS, LLC, USA). In a prospective, multicenter study of perioperative blood loss in adult patients undergoing spinal fusion surgery, the total estimated perioperative blood loss was approximately 25.5% lower in

patients receiving TXA compared to versus placebo (1592 mL vs. 2138 mL, p = 0.026) [54]. In another randomized controlled trial in elderly patients undergoing hip fracture surgeries, patients who received TXA administration had a 26.4% reduction in mean intraoperative blood loss (902.4 mL vs. 1226.3 mL; p = 0.003) [55]. Only a few research focusing on neurosurgery reported the intraoperative hemorrhage volume for meningioma resection. In a prospective study of blood loss in neurosurgical patients, TXA significantly reduced intraoperative blood loss by 24.6% (817 mL vs. 1084 mL; p = 0.012) [7]. In addition, a systematic review of the use of TXA for elective resection of intracranial neoplasms demonstrated that TXA significantly reduced intraoperative blood loss by 25.2% (821.9 mL vs. 1099.0 mL; p < 0.05) [56]. On the other hand, in small randomized controlled trials, patients undergoing meningioma resection who received TXA administration had a 26% to 46% reduction in intraoperative blood loss with baseline blood loss of around 1100 mL [4, 57]. Therefore, We assume that the estimated blood loss in huge meningioma resection will be 1000 mL, and TXA could reduce intraoperative blood loss by 25%, with a within-group standard deviation of 400 mL [7, 29]. A sample size of 228 patients (76 per group) is calculated to provide 90% power to detect the between-group difference at a significance level α of 0.017 and the overall type I error of 0.05, with a dropout rate of 2.5% with Bonferroni adjustment.

### Data monitoring committee

A Data Monitoring Committee (DMC), composed of specialists in neurosurgery, anesthesiology, ethics, methodology and statistics will monitor the trial. The DMC will audit through regular interviews or telephone calls. The DMC is also responsible for terminating the research in case of serious adverse events.

### Reporting of adverse events

The adverse effect of TXA will be closely monitored from the start of infusion to discharge. All adverse events associated with the trial will be recorded, treated, and closely monitored until a stable situation or stabilization has been reached. The serious adverse events will be informed to the principal investigator and the principal investigator will determine the severity and causality. The severity and causality of all adverse events associated with the study will be recorded and reported to the ethics committee within 24 hours. Responsible anesthesiologists will record all the adverse effects, including the type, the diagnosis time, the duration, treatment, and the consequences, and they have an obligation to stop the infusion of the study agent based on their decision.

### Emergency unblinding

We expect the need for emergency unblinding to be relatively rare. Nevertheless, we have the following procedure in case emergency unblinding is required:

If unblinding is deemed to be necessary for the intraoperative event of significant safety concerns, the principal investigator will be notified immediately. If the principal investigator considers emergency unblinding necessary, a request will be directed to the responsible anesthesiologist. The requested information and details of emergency unblinding will be recorded in the case report form and well-documented. The actual allocation must not be disclosed to the participant or other trial personnel since the trial is placebo controlled-double-blind.

### Data management

All collected data will be kept strictly confidential for research purposes only. Documents are stored safely in confidential conditions. Subject names or other identifiable data are not

included in any reports, publications, or other disclosures, except where required by law. A paper case report form and an electronic data collection form (EpiData version 4.6, EpiData Association) will be used simultaneously. Files containing information from the participants will be stored in a locked filing cabinet. Electronic data will be stored in a research computer and are coded and stored in encrypted files that required password entry, and only the principal investigators and the statistician had full access to the data content. All data will be monitored and reviewed by the principal investigator. Data backup will be performed on a quarter basis on another external hard drive.

## Protocol amendment

The principal investigator will be responsible for any decision to amend the protocol. If there is any modification (e.g., changes to eligibility criteria, outcomes, analysis), the principal investigator will communicate and gain approval from the Ethics Committee of Beijing Tiantan Hospital, Capital Medical University, prior to implementation. The principal investigator will also be responsible for communicating with relevant other parties.

## Ethics and dissemination

The Ethics Committee of Beijing Tiantan Hospital of Capital Medical University (ky2022-075-02) approved the protocol (V.1.1.1, Jan 2022). The findings of this study will be disseminated in peer-reviewed journals and at scientific conferences.

## Discussion

We present the rationale and study protocol for this randomized, placebo-controlled, and double-blinded trial, which is aiming to investigate the effect of intraoperative TXA infusion on the intraoperative blood loss of huge meningioma resection. Our hypothesis is that intraoperative administration of different doses of TXA can effectively reduce intraoperative bleeding in patients undergoing resection of huge meningioma.

The effect of intravenous TXA for reducing intraoperative blood loss was repeatedly reported, yet the optimal dose for intracranial neoplasm populations with high risk of excessive hemorrhage was controversial. Several trials have shown a greater reduction in blood loss with high doses than with lower ones in cardiac surgeries [58, 59]. A prospective interventional dose-dependent study reported that intraoperative low-dose TXA (10 mg/kg followed by 1 mg/kg/hour over 12 h) infusion effectively reduced 36% intraoperative blood loss (365 mL vs. 552 mL; p = 0.042) in patients undergoing cardiac operation with extracorporeal circulation [60]. On the other hand, intraoperative high-dose TXA may contribute to an increased risk of postoperative seizure [30, 61–63]. A recent meta-analysis showed that intraoperative high-dose TXA infusion was associated with an increased incidence rate of postoperative seizure (incidence rate: 5.3%; 95%CI: 3.3%-7.3%; $I^2$ = 56%; p < 0.001) in cardiac surgery [62]. Three trials in neurosurgery focused on the effect of intravenous TXA administration [4, 7, 36]. The loading doses ranged from 10 to 25 mg/kg, administered either as a bolus [7, 36] or a 20-minute infusion [4]. The continuous infusion rate varied from 1 mg/kg/h [4, 7] to 5–10 mg/kg/h [36]. In the retrospective review of 519 patients undergoing complex skull base surgeries, TXA was associated with reduced blood transfusion rate (7% vs. 13%, p = 0.04) with no apparent increase in thrombotic complications (risk difference: -0.9%; 95%CI: -3.5% to 1.8%) or seizure (risk difference: 1.1%, 95%CI: -1.7% to 3.9%) [36]. In a randomized controlled trial with 60 patients undergoing meningioma resection, lower intraoperative blood loss was demonstrated (830 mL vs. 1124 mL; p = 0.03) in the TXA group, while the incidence of postoperative new neurologic deficit was comparable in the groups (13.8% vs. 33.3%, p = 0.12) [4]. Vel et al.

demonstrated in their randomized controlled study that there was a significant reduction in the volume of intraoperative blood loss in the TXA group (817 mL vs. 1084 mL, p = 0.012) among patients undergoing craniotomy for tumor excision [7]. In conclusion, the dose used in the above studies reduced intraoperative blood loss and the intraoperative transfusion rate without increasing the incidence of TXA-associated complications. To ensure the relative stability of intraoperative TXA plasma concentration and test whether the hemostatic effect of TXA increases in a dose-dependent manner, continuous TXA infusion is adopted until dura closure. Therefore, our study adopted a bolus of 20 mg/kg followed by continuous infusion at a rate of 5 mg/kg/h.

Huge meningiomas with a diameter ≥ 5cm may lead to an intracranial occupying effect and often involve the surrounding dura structures, including major venous sinus, cavernous sinus, arteries, and cranial nerves. The involvement with important intracranial structures may contribute to obstruction of venous drainage, thus resulting in brain edema [64]. Moreover, the excessive turbulence of hemodynamic caused by sinus and artery laceration will also increase the incidence of artery spasm, postoperative stroke and cardiac event. Therefore, the strategy for intraoperative blood management for huge meningioma is essential in perioperative care.

The primary outcome of this study is the estimated blood loss, which is calculated based on the volume of suction bottle, sponge from the operative field, and flushing. This is the most commonly used method in intraoperative blood loss estimation and can be performed directly in the operating room without additional equipment [49]. However, the estimated blood loss may be challenging in case of significant blood loss or blood transfusions [50]. We also calculated intraoperative CBL of each patient using the method described by López-Picado A as supplementary evidence to detect the blood-sparing effect of TXA [51]. CBL can provide an additional reference for intraoperative blood loss estimation. By measuring both estimated blood loss and CBL, we can understand the effect of TXA more accurately and comprehensively [51].

This study will not exclude patients with a history of epilepsy. Currently, the association between postoperative seizure and TXA was mainly reported in cardiac, trauma, and orthopedic [63]. Few studies reported the association of TXA and postoperative seizure in the neurosurgical population. To minimize possible risks, standardized postoperative anti-epileptic treatment will be performed [65]. Furthermore, relatively low dose are selected in this trial.

The present study is a parallel-group design study with explicit target population, well-established randomized and blind settings, and a rigorous uniform protocol to perform standard perioperative management. Furthermore, experienced anesthesiologists, neurosurgeons, and neurologists are strictly following study protocol in each essential procedure in intervention and follow-up. This is one of the early studies focusing on neurosurgical patients with a high risk of massive bleeding to evaluate the blood-sparing effect of intraoperative infusion of TXA.

Our study will provide reference to optimize perioperative blood management during huge meningioma resection, thus improving perioperative management of patients at high risk of massive hemorrhage.

## Supporting information

**S1 File. SPIRIT checklist.**
(DOC)

**S2 File. Research project submitted to the ethics committee (English).**
(DOC)

**S3 File. Research project submitted to the ethics committee (Chinese).**
(DOC)

**S1 Table. Definition of complications.**
(DOCX)

## Author Contributions

**Conceptualization:** Haojie Yu, Minying Liu, Shu Li.

**Data curation:** Shu Li.

**Funding acquisition:** Shu Li.

**Methodology:** Haojie Yu, Minying Liu, Shu Li.

**Project administration:** Haojie Yu, Minying Liu, Xingyue Zhang, Tingting Ma, Jingchao Yang, Yaru Wu, Jie Wang, Muhan Li, Juan Wang, Min Zeng, Liyong Zhang, Hailong Jin, Xiaoyuan Liu, Shu Li, Yuming Peng.

**Supervision:** Shu Li.

**Writing – original draft:** Haojie Yu, Minying Liu, Shu Li.

**Writing – review & editing:** Haojie Yu, Minying Liu, Xingyue Zhang, Tingting Ma, Jingchao Yang, Yaru Wu, Jie Wang, Muhan Li, Juan Wang, Min Zeng, Liyong Zhang, Hailong Jin, Xiaoyuan Liu, Shu Li, Yuming Peng.

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
