## [Decision Letter · Decision Letter 0]

3 Apr 2023

PONE-D-23-00579

The effect of tranexamic acid on intraoperative blood loss in patients undergoing brain meningioma resections: study protocol for a randomized controlled trial

PLOS ONE

Dear Dr. Li,

Thank you for submitting your manuscript to PLOS ONE. After careful consideration, we feel that it has merit but does not fully meet PLOS ONE’s publication criteria as it currently stands. Therefore, we invite you to submit a revised version of the manuscript that addresses the points raised during the review process.

ACADEMIC EDITOR: 

The present manuscript is an actual protocol and not a final study with its results, and although its very good and impressive yet It would seem more convenient to add any pilot data available to give an idea about the course of the study and its expected duration to be finalized  and to have the results also published.It is required to conduct a final English review because of the presence of some grammatical errors that need to be addressed.Based on my review, I didn't understand the secondary outcome,,, you indicated that you will calculate the blood loss, does that mean it's a postoperative blood loss assessment?, and if so then for how long are you calculating this losses. And why didn't this be related (compared ) to the primary outcome. OR is it intra-operative too and is so then what is the difference between the assessment of blood loss in the primary and the secondary outcomes.

please answer the reviewers comments.

We look forward to receiving your revised manuscript.

Kind regards,

Hossam Eldien Ahmed Anis ElShamaa, M.D.

Academic Editor

PLOS ONE

Journal Requirements:

Reviewers' comments:

Reviewer's Responses to Questions

**Comments to the Author**

1. Does the manuscript provide a valid rationale for the proposed study, with clearly identified and justified research questions?

Reviewer #1: Yes

Reviewer #2: Partly

2. Is the protocol technically sound and planned in a manner that will lead to a meaningful outcome and allow testing the stated hypotheses?

Reviewer #1: Yes

Reviewer #2: Partly

3. Is the methodology feasible and described in sufficient detail to allow the work to be replicable?

Reviewer #1: Yes

Reviewer #2: No

4. Have the authors described where all data underlying the findings will be made available when the study is complete?

Reviewer #1: Yes

Reviewer #2: Yes

5. Is the manuscript presented in an intelligible fashion and written in standard English?

Reviewer #1: No

Reviewer #2: Yes

6. Review Comments to the Author

You may also provide optional suggestions and comments to authors that they might find helpful in planning their study.

Reviewer #1: In manuscript PONE-D-23-00579, Li and co-authors present their research on the effect of tranexamic acid (TXA) on intraoperative blood loss in patients undergoing brain meningioma resections. Unfortunately, the manuscript submitted is not a completed research protocol, but instead is they present the basis for their study protocol only. In other words, no patients have been enrolled and no data/results are presented. I admit that the protocol, including the reasons for their study based on prior work, is compelling and appears to be quite comprehensive. However, without any data, it cannot be published.

A few comments for the authors:

1. The manuscript should be reviewed by a native English writer as many grammatical errors, complex sentence structure, and other issues can be found throughout the paper. Additionally, the paper is written in future tense (my first clue that the paper was not a research study) and would need to be rewritten in the more common past tense.

2. The primary and secondary outcomes both include blood loss. For the primary outcome, estimated intraoperative blood loss was calculated based on blood waste seen, whereas the secondary outcome includes calculated intraoperative blood loss. The authors do describe the differences in their methods; however, I am not sure why they need to measure both.

3. The authors often use qualitative terms without defining them. For example, “huge meningioma” resection should be described as meningiomas > 5 cm. Additionally, “Massive intraoperative bleeding” (page 11 line 79) should be defined. What is massive, > 500 ml, 1000 ml?

4. Why will ASA IV patients not be included in this study?

Reviewer #2: My main concerns with this study is, that tranexamic acid can cause seizure and cerebral artery constriction and this issue has not been considered at all. What happens with patients with preoperative epilepsy or prevoius seizures?

2. The type of anesthesia is not standardized. Therefore, etomidate and propofol are not equivalent.

3. I have not found sufficient preoperative variables to compare the study arms.

7. PLOS authors have the option to publish the peer review history of their article (what does this mean?). If published, this will include your full peer review and any attached files.

Reviewer #1: **Yes: **Timothy Angelotti MD PhD

Reviewer #2: No

---

## [Author Response · Author response to Decision Letter 0]

15 May 2023

Response to Reviewers

Reviewer 1

In manuscript PONE-D-23-00579, Li and co-authors present their research on the effect of tranexamic acid (TXA) on intraoperative blood loss in patients undergoing brain meningioma resections. Unfortunately, the manuscript submitted is not a completed research protocol, but instead is they present the basis for their study protocol only. In other words, no patients have been enrolled and no data/results are presented. I admit that the protocol, including the reasons for their study based on prior work, is compelling and appears to be quite comprehensive. However, without any data, it cannot be published.

RESPONSE: Thank you. This study protocol is presented following the Standard Protocol Items: Recommendations for Interventional Trial (SPIRIT) statement, and the SPIRIT checklist. Trial data or results are not recommended to be reported in the protocol. At the time of manuscript submission, the study is still recruiting participants. We enrolled the first patient on 15th June 2022, 160 patients were recruited at the time of submission, and we expect to recruit 228 patients and complete the study by Dec 2023. Detailed information on the study status is presented in the trial status section. Please see page 24 line 475

1. The manuscript should be reviewed by a native English writer as many grammatical errors, complex sentence structure, and other issues can be found throughout the paper. Additionally, the paper is written in future tense (my first clue that the paper was not a research study) and would need to be rewritten in the more common past tense.

RESPONSE: Thank you. We’ve revised the manuscript with an authentic speaker and corrected grammatical errors. Clinical trial protocols should be presented in the future tense as the study design is established before the initiation of the study. Most trial protocols are presented in the future tense. Please see below study protocol for reference: Zeigelboim BS, Jose´ MR, Santos GJBd, Severiano MIR, Teive HAG, Stechman-Neto J, et al. (2021) Balance rehabilitation with a virtual reality protocol for patients with hereditary spastic paraplegia: Protocol for a clinical trial. PLoS ONE 16(4): e0249095. https://doi.org/10.1371/journal. pone.0249095.

2. The primary and secondary outcomes both include blood loss. For the primary outcome, estimated intraoperative blood loss was calculated based on blood waste seen, whereas the secondary outcome includes calculated intraoperative blood loss. The authors do describe the differences in their methods; however, I am not sure why they need to measure both.

RESPONSE: Thank you. The primary outcome is the estimated intraoperative blood loss, which is assessed based on the volume of the suction bottle, sponge from the operative field, and flushing. This assessment of blood loss is the most used method and can be performed directly in the operating room without additional equipment. The accuracy of this assessment is challenging, particularly in case of massive blood loss and blood transfusions. On the other hand, calculated blood loos is drawn from the formula that calculated blood loos = estimated blood volume × (Hcti−Hctf) + transfused RBC volume] / Hctmean. In this formula, Hcti is the initial hematocrit at the beginning of surgery, Hctf is the final hematocrit at closure, and Hctmean is the mean hematocrit (between initial and final). By measuring both estimated blood loss and calculated blood loss, we can understand the effect of TXA more accurately and comprehensively. We revised the manuscript to further clarify the difference between the primary outcome and calculated blood loss. Please see page 22 line 440.

3. The authors often use qualitative terms without defining them. For example, “huge meningioma” resection should be described as meningiomas > 5 cm. Additionally, “Massive intraoperative bleeding” (page 11 line 79) should be defined. What is massive, > 500 ml, 1000 ml?

RESPONSE: Thank you. We add the definition of huge meningioma as well as the definition of massive intraoperative bleeding in the manuscript. For the secondary outcome. We also defined massive hemorrhage in the S1 Table. Please see page 4 line 69, page 4 line 79, and S1 Table.

4. Why will ASA IV patients not be included in this study?

RESPONSE: Thank you. According to the American Society of Anesthesiologists (ASA) physical status classification system, ASA Ⅳ patients are patients with a severe systemic disease that is a constant threat to life. Example: Patient with functional limitation from severe, life-threatening disease (e.g., unstable angina, poorly controlled COPD, symptomatic CHF, recent (less than three months ago) myocardial infarction or stroke). These patients are extremely fragile and vulnerable, and it may be unethical to conduct a clinical trial on these patients. Therefore, we excluded ASA IV patients.

Reviewer 2

1. My main concerns with this study is, that tranexamic acid can cause seizure and cerebral artery constriction and this issue has not been considered at all. What happens with patients with preoperative epilepsy or previous seizures?

RESPONSE: Thank you. Currently, the association between postoperative seizure and TXA was mainly reported in cardiac and vascular surgery. In neurosurgery, few research reported the association. Our previous work, which was trying to detect the effect of TXA on seizure-free meningioma patients regard to postoperative seizure, indicated that a low-dose TXA would not increase the risk of seizure in these populations (4.3% vs 3.7%, risk difference, 0.7%; 1-sided 97.5% CI, -∞ to 4.3%; P = 0.001 for noninferiority). This work is not published yet. Patients with preoperative epilepsy undergoing neurosurgery are at high risk of postoperative seizure, but again, no literature reports the association of TXA with seizure in this population. In our study, we select relatively lower doses of TXA both in the low-dose group and high-dose group, compare to cardiac surgeries, to avoid the possible increased risk of seizure. Furthermore, the postoperative seizure will be closely monitored, and all participants will follow standardized postoperative anti-epileptic treatment. We also revised the manuscript accordingly. Please see page 23 line 450.

2. The type of anesthesia is not standardized. Therefore, etomidate and propofol are not equivalent.

RESPONSE: Thank you. Etomidate and propofol are both sedatives for anesthesia induction and there is no evidence suggesting any of these two drugs in anesthesia induction would affect the measurement of intraoperative blood loss. To standardize anesthesia management, all participants will receive combined intravenous and inhalational anesthesia.

3. I have not found sufficient preoperative variables to compare the study arms.

RESPONSE: Thank you. We presented the preoperative evaluations in Table 1 and presented in the manuscript. We revised the manuscript accordingly to clarify our data collection and measurement related to these variables. Please see page 12 line 223.

---

## [Editor Report · Decision Letter 1]

15 Aug 2023

The effect of tranexamic acid on intraoperative blood loss in patients undergoing brain meningioma resections: study protocol for a randomized controlled trial

PONE-D-23-00579R1

Dear Dr. Shu,

We’re pleased to inform you that your manuscript has been judged scientifically suitable for publication and will be formally accepted for publication once it meets all outstanding technical requirements.

Kind regards,

Hossam Eldien Ahmed Anis ElShamaa, M.D.

Academic Editor

PLOS ONE

---

## [Editor Report · Acceptance letter]

21 Aug 2023

PONE-D-23-00579R1 

The effect of tranexamic acid on intraoperative blood loss in patients undergoing brain meningioma resections: study protocol for a randomized controlled trial 

Dear Dr. Li:

I'm pleased to inform you that your manuscript has been deemed suitable for publication in PLOS ONE. Congratulations! Your manuscript is now with our production department. 

Kind regards, 

on behalf of

Dr. Hossam Eldien Ahmed Anis ElShamaa 

Academic Editor

PLOS ONE